# Therapeutic Monoclonal Antibodies against Cancer: Present and Future

**DOI:** 10.3390/cells12242837

**Published:** 2023-12-14

**Authors:** Marisa Delgado, Jose A. Garcia-Sanz

**Affiliations:** Department of Molecular Biomedicine, Centro de Investigaciones Biológicas Margarita Salas (CIB-CSIC), 28040 Madrid, Spain; marisadelgado@cib.csic.es

**Keywords:** therapeutic antibodies, cancer treatment, transmembrane proteins, bioinformatics, cancer database analyses

## Abstract

A series of monoclonal antibodies with therapeutic potential against cancer have been generated and developed. Ninety-one are currently used in the clinics, either alone or in combination with chemotherapeutic agents or other antibodies, including immune checkpoint antibodies. These advances helped to coin the term personalized medicine or precision medicine. However, it seems evident that in addition to the current work on the analysis of mechanisms to overcome drug resistance, the use of different classes of antibodies (IgA, IgE, or IgM) instead of IgG, the engineering of the Ig molecules to increase their half-life, the acquisition of additional effector functions, or the advantages associated with the use of agonistic antibodies, to allow a broad prospective usage of precision medicine successfully, a strategy change is required. Here, we discuss our view on how these strategic changes should be implemented and consider their pros and cons using therapeutic antibodies against cancer as a model. The same strategy can be applied to therapeutic antibodies against other diseases, such as infectious or autoimmune diseases.

## 1. Introduction

The discovery in the mid-seventies of the technology to generate monoclonal antibodies (mAb) by George Köhler and Cesar Milstein [1] led to an effort to obtain new mAbs, allowing to identify new antigens for diagnostic or therapeutic purposes. mAbs were used at the beginning of the eighties as markers to localize a tumor [2], and in the mid-eighties, their therapeutic use started [3]. Over the last decades, mAbs have become indispensable tools in research, diagnostics, and therapy [4], in particular for the diagnosis and treatment of cancer [5] or inflammatory and autoimmune diseases [6,7,8], as well as for the treatment of other conditions, including migraine [9]. Table 1 shows the distribution in therapeutic areas of the 197 antibodies approved by the FDA/EMA or other medicine agencies, including some on late review. Nearly half (91) correspond to antibodies approved for cancer treatment. The approved antibodies or those in late review are analyzed yearly by publications of The Antibody Society [10,11,12,13,14,15,16,17,18,19,20,21,22,23,24]. The cumulative data on these antibodies can be accessed through The Antibody Society webpage [25]. Here, Appendix A lists each anti-cancer therapeutic antibody and describes its main characteristics. The clinical trials for each antibody are out of the scope of this review, but they are described in several recent reviews [5,26,27,28,29].

Most anti-tumoral antibodies recognize antigens present on the surface of the tumor cells. Others, rather than directed to the tumor cells themselves, recognize surface antigens interfering with neo-vascularization, inhibit migration, or target metalloprotease secretion and tumor cell invasion [5]. A third group recognizes surface antigens present in the cells of the immune system. These antigens are involved in (i) attracting the lymphoid cells to the tumor (chemokine receptors) [30]; (ii) receptors present either in the APCs or the T cells that increase the patient’s immune response after interaction with their ligands, where Abs that act as agonists for these receptors can, therefore, increase T cell activation and effector functions against the tumor [31]; and (iii) molecules delivering negative signals for the activation of the T lymphocytes (immune checkpoints). If antibodies block the interaction of the immune checkpoints with their ligands, this can switch the immune response from an immunosuppressed anti-tumoral status to effectively attack the tumor [31,32]. Therefore, current trends for therapeutic antibodies in oncology include delaying (or inhibiting) tumor growth and interfering with the generation of metastases.

The first group of antibodies discussed in more detail recognizes antigens present on the surface of the tumor cells. They may identify plasma membrane antigens or a ligand or receptor. In some cases, the ligand–receptor interaction might be necessary for the survival of the tumor cells. Therefore, blocking receptor–ligand binding and the corresponding signaling cascade can have robust therapeutic effects [33,34,35,36,37,38,39,40,41] (Figure 1A). Antibodies directed against ligands, such as MABp 1 (anti-IL-1a) or Denosumab (anti-RANK-L), have been effective since binding of the antibodies prevents the interaction of the ligands with their receptors IL1R [40] or RANK [41], respectively. However, it is generally more effective to use antibodies against the receptor, for which multiple examples have been described, including Olaratumab, an antibody recognizing the platelet-derived growth factor receptor alpha (PDGFRα) [33,34,35,36]; Cetuximab, which prevents ligand binding to EGFR and triggers internalization of the receptor and its degradation [37]; or the antibodies Pertuzumab and Trastuzumab, which recognize the HER2 receptor [30], inhibit its homo- and hetero-dimerization, and prevent HER2 signaling [38,39].

A different type of approach to cancer therapy is to increase the patient’s T cell-mediated immune response by utilizing agonist Abs recognizing some members of the TNFR family that activate T cells. These include CD40 [31,42,43], GITR [44], OX40 [31], and 4-1BB [31]. In this case, the agonistic antibodies can positively signal in the antigen-presenting cells (CD40) or on the effector T cells (OX40, 4-1BB and GITR) and burst their activation and effector functions (Figure 1B). Currently, these therapeutic approaches are being tested in clinical trials, some of them with promising results.

For antibodies interfering with neo-vascularization, inhibiting migration, or targeting metalloprotease secretion and tumor cell invasion, the aims are (i) the inhibition of new blood vessel generation either through targeting the VEGFR2 receptor (Ramucirumab) [45] or VEGF [46]; (ii) interfering with cell migration, which includes antibodies against PDGFRa [33,34,35,36], VEGFR2 [45], and VEGF [46], or cell adhesion, which includes antibodies against GD2 [47], EpCAM [48], CD33 [49], CD38 [50], or CD52 [51]; or (iii) inhibiting tumor cell invasion and metastasis formation, for which a relevant role has been described for some anti-chemokine receptor antibodies [30].

The last group of antibodies recognizes surface antigens in cells of the immune system, either to attract them to the tumor or to switch the response of these cells from an immunosuppressed anti-tumoral status to another that allows them to effectively attack the tumor (the so-called immune checkpoint antibodies). The anti-immune checkpoint antibodies are known to block the negative signals given to the host’s T cells by the interaction of the CTLA-4 receptor with the B7.1 or B7.2 antigens of the PDL-1 or PDL-2 ligands present on the tumor cells with the PD-1 receptor expressed on the T cells [32,52,53,54] (Figure 1C) or the NK checkpoints of the KIR family [55,56].

Although most therapeutic antibodies have been generated in mice, very few (~5%) are used as rodent Ig. The rest have been engineered and used either as chimeras (<10%) or more often as humanized antibodies (~50%). In the latter, the only mouse-derived sequences in the immunoglobulin (which can generate an immune response against non-self) are the complementarity-determining regions (CDR) from the light and heavy Ig chains. In addition, most humanized antibodies have been engineered into human IgG1 sequences since this isotype can activate complement and recruit immune effector cells for ADCC, whereas IgG2 does not activate complement, and IgG4 does not activate ADCC or CDC. The remaining therapeutic antibodies are of human origin, partially due to the successful generation of mice strains with a humanized Ig locus [57,58,59,60,61,62,63,64,65,66,67,68,69,70,71]. Engineering has been aiming to modify the function of the antibodies, improving FcγRIIIa binding, and ADCC and decreasing the affinity for the human inhibitory FcγRIIB (CD32B) [72]; stabilizing the hinge; extending the half-life of the antibody in vivo; or even removing Fc effector functions (ADCC/ADCP) [25]. In addition, other Ig classes (IgA, IgE, or IgM) are being evaluated as anti-cancer therapeutic antibodies. For example, the therapeutic possibilities of an IgE-based immunotherapy [69,73,74,75,76], including its safety [71,77], have been analyzed in renal carcinoma [67], breast cancer [68], ovarian cancer [78], and melanoma [79] models. All these analyses showed that the IgE was more effective than the corresponding IgG1 antibody. In addition, IgA antibodies can lead to increased tumor killing by neutrophils [59,60,61,62,63,64,65], suggesting a novel anti-cancer strategy. Finally, there are also analyses of the IgM antibodies [66], where their pentameric structure allows the possibility of combining several cytokines with bispecific or trispecific antibodies.

Different mechanisms of action might be responsible for a delay in tumor growth: Some antibodies (i.e., anti-CD95 mAb or bispecific anti-CD95) directly target apoptosis receptors and kill the tumor cells through apoptosis [80,81]. Others may recognize antigens not classically associated with the induction of cell death, like Dinutuximab (anti-GD2), but which also induce direct tumor cell death [47]. The vast majority of therapeutic antibodies induce tumor cell death through the interaction with other molecules or cells in the host’s immune system by activation of antibody-dependent cell-mediated cytotoxicity (ADCC), like Cetuximab [37]; by activation of antibody-dependent cell phagocytosis (ADCP), like Trastuzumab [82]; or by triggering complement-mediated cytotoxicity (CDC), like Edrecolomab [48]. However, many therapeutic antibodies to exert their functions, rather than using a single mechanism of action, might use several of them combined. For example, Edrecolomab (anti-EpCAM) uses a combination of CDC, ADCC, and ADCP to inhibit tumor growth [48], whereas Dinutuximab combines apoptosis, CDC, and ADCC [47] (Figure 1). Other antibodies have been engineered to recognize a different antigen in each arm of the antibody (bispecific antibodies) to be conjugated to drugs (antibody-drug conjugates or ADC) as immunoconjugates, as radioactive-isotope conjugated antibodies, or conjugated with photoimmunotherapeutic agents. In addition, the antibodies can be employed either alone or, more often, in combination with other antibodies or other therapies, including chemotherapy, radiotherapy, molecular inhibitors, etc. [5]. Our experience with an anti-CCR9 antibody has demonstrated that a combination of suboptimal doses of the antibody in combination with suboptimal doses of vincristine led to a synergistic increase in survival of the xenotransplanted mice, where 40% of them survived >440 days with undetectable tumor cells in the spleen, bone marrow, or liver [83]. Many other examples of synergistic effects between antibody therapy and chemotherapy have been reported [84,85,86,87,88,89,90], including immune checkpoint antibodies with chemotherapeutic agents [91] but also antibodies in combination with inhibitors [92] or with dendritic cell vaccines [93]. In general, the affinities of the therapeutic antibodies are in the nanomolar range, and many cases are genetically modified to generate second- and third-generation antibodies with enhanced effector functions.

## 2. An Example: HER2 Targeting

HER2, a member of the ErbB family of receptor tyrosine kinases, is over-expressed in approximately 25% of human breast cancers, giving its name to a breast cancer subtype [94]. HER2 over-expression was associated with poor patient outcomes until the development of HER2-targeted therapies [95,96]. Trastuzumab is a therapeutic antibody against HER2 that has shown significant clinical benefit, including a 50% reduction in the risk of death after treatment, concomitant with a similar improvement in disease-free survival [82,97,98,99]. However, it is worth mentioning that a significant fraction of HER2^+^ breast cancer patients, particularly the ones with advanced HER2 gene amplified breast cancer, after treatment with anti-HER2 antibodies eventually relapse or develop progressive disease [100], suggesting that tumors either possess or acquire intrinsic mechanisms of resistance allowing escape from HER2 inhibition mechanisms [100,101,102]. HER2 over-expression is not restricted to a subset of mammary gland tumors since it is also over-expressed in mammary tumors from other subtypes, like basal, luminal A, or luminal B breast tumors (Figure 2A), as well as in a small fraction of other tumor types (Figure 2B). Thus, all patients with high levels of HER2 expression, independently of the breast cancer subtype to which they were classified, can benefit from targeted anti-HER2 therapies, as well as any patients with another type of solid tumor over-expressing HER2. Indeed, trastuzumab is being used for therapeutic purposes in some metastatic colorectal cancers [103], uterine papillary serous carcinomas [104], pancreatic adenocarcinomas [105], metastatic biliary tract cancers [106], vulvar Paget’s disease [107], and epithelial ovarian cancers [108]. However, HER2 is also expressed in some tumors from the uterus, urothelium, thyroid, stomach, skin, kidney, head and neck, esophagus, bladder, and hematopoietic tumors (Figure 2B), where antibodies against HER2 could also be used for therapeutic purposes.

In addition, patients with breast tumors with high HER2 mRNA expression levels have a 26% reduced survival compared to patients with breast tumors with low/intermediate HER2 expression levels (126.37 ± 3.1 months for low/intermediate expression versus 99.96 ± 6.6 months for high HER2-expressing tumors) (Figure 3A). These survival differences show the relevance of HER2 gene over-expression on patient outcomes. Furthermore, HER2 expression is highly restricted in normal tissues since only samples from 6 patients expressed high HER2 levels (five of normal breasts and one stomach sample) (Figure 3B). Thus, these data confirm the potential of HER2 as a therapeutic target.

## 3. Future

The role of monoclonal antibodies as therapeutic agents for cancer treatment has been widely demonstrated [5]. It turns out that in many cases of patients treated with therapeutic antibodies, similar to what was described above for patients carrying HER2^+^ tumors and treated with anti-HER2 antibodies, patients eventually relapse or develop progressive disease [100], suggesting that tumors either possess or acquire intrinsic resistance mechanisms, allowing tumor escape [100,101,102]. Drug resistance is a relevant challenge for cancer treatment, not restricted to chemotherapeutic drugs, since it also affects antibody-based therapies [29,110]. Tumors treated with antibodies either evolve by bypassing the signaling associated with that particular receptor [5,29,111] or towards variants not expressing the target antigen. Thus, in the future, the goal of anti-tumor immune therapies should be to trigger from the beginning all possible host defense mechanisms, aiming to destroy as early as possible the highest number of tumor cells. This strategy will decrease the probability of the tumor developing escape mechanisms and consequently increase the effectiveness of these therapies, thus, raising the following question: How can this be carried out more effectively in the near future?

We believe there should be strategic changes in therapeutic antibody generation for cancer treatment to make it more efficient. The goal will be to attack each tumor, rather than with a single therapeutic antibody recognizing antigens in the surface of the tumor cell, as is the case for most tumors (i.e., HER2^+^ tumors treated with trastuzumab, pertuzumab, or their derivatives), with an appropriate combination of antibodies, preferentially recognizing different antigens or different epitopes of the same antigen expressed on the tumor cell surface, as initially demonstrated with the combination of trastuzumab and pertuzumab with docetaxel for the treatment of patients with HER2-amplified metastatic breast cancer [112]. This approach would increase the probability of destroying the highest number of tumor cells as early as possible, decreasing, as a consequence, the possibilities of the tumor developing escape mechanisms. For this purpose, large panels of monoclonal antibodies should be generated. Samples from each patient will be tested for the antibody panel to determine the set or antibodies that recognize antigens expressed, or over-expressed, in the tumor cells of that particular patient and could be used in combinations from the initial treatment to promote an amplification of the anti-tumor immunotherapeutic response.

To undertake such an effort, a large multidisciplinary consortium, rather than a single research group, will be required, together with the appropriate funding.

We propose the selection of antigens for the immunizations, using an “educated guess” based on the over-expression of an mRNA in the tumor, compared to the normal tissue, while discarding genes widely expressed in other normal tissues. This decision is based on the analysis of the GENT2 database (>60,000 human samples) [109] or equivalent databases such as the Center for Cancer Genomics (TCGA) [113] or the cBioportal [114,115,116]. In the following paragraphs, we aim to discuss the strategies in more detail, including the choice of tumor type, the requirements of the proteins as possible therapeutic targets, the selection of the immunogens, the generation of the antibodies, and the screening methods. Pitfalls and alternatives are discussed throughout the text.

### 3.1. Choice of the Tumor Type

The first criterion will be to decide the tumor type on which to concentrate the efforts. For this purpose, the incidence (Figure 4A) and mortality (Figure 4B) rates for different tumor types [117] might be valuable since they classify the types of tumors based on the number of new cases or mortality, respectively. However, other parameters, such as the mortality-to-incidence ratio (Figure 5), might seem more appropriate since they allow the classification of the different tumors based on their life-threatening activity. It shows, for example, that two types of tumors with a similar incidence, like thyroid tumors (>5 × 10^5^ new cases/year, 9th on the incidence list) and pancreatic cancer (~5 × 10^5^ new cases/year, 12th on the incidence list), have contrasting outcomes. Whereas >90% of the patients with thyroid tumors survive, >90% of the patients diagnosed with pancreatic cancer die. These data are available from public databases. In the examples shown here, the data was obtained from the International Agency for Research on Cancer from the World Health Organization [118] but are also available from other sources, such as the American Cancer Society [119] or Cancer Research UK [120].

### 3.2. Analysis of Cell Surface Proteins

The second criterion is that the antigen should be expressed in the cell surface to be recognized by the antibody. Although some effort has been made to generate therapeutic antibodies against intracellular tumor antigens [122,123], the vast majority of the therapeutic antibodies approved or under development identify proteins present on the surface of the cells. Thus, for the initial analysis of the possible protein targets expressed by a given type of tumor, instead of analyzing all the protein-coding genes, the analysis can be restricted to cell surface proteins, which represent between 24 and 30% of the proteins coded by the human genome [124,125,126,127], which was estimated to be 5539 human genes that code for cell membrane proteins [127]. This estimate corresponds very well to the 5591 gene entries found in the gene ontology databases [128,129,130] under the search term GO:0005886 (plasma membrane). Proteins in the cell surface can be identified by other bioinformatic analyses [113,114,115,116] or artificial intelligence [131]. This second criterion reduces the complexity of the genes entering the analyses and concomitantly the amount of work on selecting the targets and increasing the probability of identifying proteins that can be antibody targets.

### 3.3. Choice of Target Genes

The third criterion is to identify genes coding for proteins in the cell surface with mRNA expression changes on tumor versus normal tissue samples. Although from the point of view of gene regulation, both over-expressed and repressed mRNAs would be interesting, the aim of generating antibodies that recognize the antigens on tumor cells restricts the analysis of over-expressed proteins on tumor samples as compared to the normal tissue counterparts. The data are accessible from public databases, such as the Human Protein Atlas [132], where a series of tools are available for analyzing the human secretome [133]; expression profiles based on antibodies [134,135], as a tool for pathology [136] or the analysis of specific cell types or tissues [132,137]; or other public databases, such as GENT2 [109], where a large number of tumor samples and normal tissues are available (52,863 tumor samples and 10,168 samples from normal tissues are available). Analysis of the Human Protein Atlas allowed to pinpoint CCR9 as a potential immunotherapeutic target for T-ALL [138], and analysis of GENT2 allowed to determine the specificity of CCR9 expression in normal and tumor tissue samples [83]. It is also relevant to check for the expression of each candidate on peripheral blood and other tissues. In particular, broadly expressed antigens would represent less desirable targets (or even bad candidates) than antigens with a more restricted expression. The rationale is that antibodies recognizing broadly expressed antigens might attack all the normal cells expressing the antigen. Thus, the aim is to identify candidates with a restricted expression pattern in healthy tissues.

Another relevant criterion is whether over-expression of the particular protein affects patient survival. For the HER2 example mentioned above, breast cancer patients with high expression levels of HER2 have a 26% decreased survival as compared to patients with low/intermediate levels of expression (Figure 3A). We have recently used a similar type of analysis to determine that patient survival negatively correlated with increased CCR9 mRNA levels [83]. Survival differences between patients with tumors expressing high or low levels of a particular gene indicate the relevance of over-expression of that gene on patient outcome. The lack of survival differences does not have any negative indication of gene relevance. These data, with information on the protein biology/biochemistry might help us decide the candidate proteins to use as targets for the antibodies.

### 3.4. Antibody Generation Strategies

Animals can be immunized with known antigens using several strategies. The first would be to subclone the cDNA coding for the desired antigen into an eukaryotic expression vector, bind the plasmid to gold nanoparticles, and use a biolistic particle delivery (gene gun) for the particle-mediated DNA immunization [139,140]. This procedure allowed generating therapeutic antibodies against CCR9 [141,142]. A second possibility would be to transfect the antigen of interest, cloned on an expression vector, on a cell line syngeneic with the mouse strain to be immunized. Since the cell line is syngeneic to the mouse strain used for the immunizations, the immune response generated will be specific to the protein coded for by the transfected cDNA. A third possibility is to immunize with synthetic peptides, an approach that allowed to obtain antibodies against CCR9 [143]. With this approach, mainly antibodies recognizing a linear peptide will be obtained. Upon immunization of an animal with an antigen, the B cells that produce antibodies recognizing that particular antigen become activated, proliferate, increase their frequency, and facilitate their selection. The hybridoma technology allows the immortalization of the immunoglobulin-secreting B cells, producing large amounts of the desired antibody over time.

Once the therapeutic potential of these antibodies is confirmed, since they are of mouse origin, they will have to be humanized. Humanization will prevent the generation by the patients of antibodies against mouse Ig, which in most cases would impair the effect of the therapeutic antibody. For this purpose, the three CDR regions from the light and heavy Ig chains are subcloned into the human germline framework sequences for IGH and IGK genes, which are subsequently fused to the constant IgG1/k immunoglobulin regions and then cloned on an expression vector that will lead to the secretion of single-chain antibodies [83]. An overview of the design and challenges has been recently reviewed [144].

The fourth possibility is to obtain directly human antibodies. The technology uses human IgG transgenic mice [145] or Ig humanized mice [57,58], immunized with the antigen of interest. Conversely, library display technologies, a methodology initially described by Richard Lerner’s group, to generate a library of human antibodies fused to M13 proteins [146], or alternative display systems, including yeast, *E. coli*, and mammalian cells, enable the selection of human antibodies of interest. In addition, methodologies to generate libraries of IgV_H_ and IgV_L_ chains that can be combined and screened for the antibodies of interest [147,148,149] allowed the identification of therapeutic antibodies for cancer treatment [150]. Other methodologies employ in vitro transcription–translation reactions, ribosome display, or mRNA display that correlate an antibody to its mRNA. These technologies and the appropriate protocols are described by Zoguska et al. [151]. The advantage of these technologies is that antibody humanization is not required. However, the lead identification phase for the display technologies allows in general the selection of low-affinity antibodies and requires a lead optimization phase in which improved variants should be generated and selected [151].

Other possibilities should be considered, like immunizing the mice with human tumor cells expressing the appropriate antigens as immunogens. However, this strategy is undesirable when the aim is to generate antibodies against a given antigen since the number of antibodies against other antigens from the surface of the cells used as immunogen can be large, and antigen identification is quite laborious. However, it allows the identification of proteins that undergo differential post-translational processing (i.e., glycosylation) in tumors and normal cells.

### 3.5. Antibody Screening

Since the antibodies recognize cell surface proteins, the best screening option is flow cytometry. Optimally, the first screening would use mouse cells stably transfected with the antigen of interest. The same cells transfected with the empty vector will represent the negative control. This screen will allow a person to select antibodies recognizing the antigen of interest. Subsequent screens will use cell lines that express the antigen of interest and several primary tumor cells. This secondary screening will allow us to verify that the antibody recognizes the endogenous antigen and the wide spread of the antigen in a panel of primary tumors. Additional analyses could be carried out with human peripheral blood mononuclear cells to verify that the antigen is either expressed at low levels or not at all in peripheral blood cells. These experiments will corroborate the expression data from the in silico analyses made with public database data.

The antibodies will be subsequently functionally tested to demonstrate whether they can inhibit or delay tumor growth. For this purpose, immunocompromised mice will be injected with appropriate tumor cell lines subcutaneously or orthotopically to generate xenotransplants. The mice used for these experiments can be either animals lacking T and B lymphocytes, such as *NOD/Scid*, *Rag 1*^−/−^, *Rag 2*^−/−^, or, preferably, can be mice that in addition to lacking T and B lymphocytes also lack mature NK cells and have a mutation on the *C5* complement gene, such as *NSG* (*NOD/Scid/IL2Recγ*^−/−^) or equivalent. Survival increases in the antibody-treated group versus the isotype control-treated group on the xenotransplanted mice carrying human model tumor cell lines or primary human tumors will determine the effectiveness of the antibody being tested with groups of animals treated with an isotype control antibody treatment, giving key answers to the therapeutic potential of each of these antibodies.

Antibodies that are not able to increase the survival of the xenotransplanted animals could still be used either to generate CAR-T cells with the antibody variable regions or be used as antibody–drug conjugates (ADC) after linking them to chemotherapeutic drugs, toxins, radioelements, etc., where the antibody directs the drug/radioelement towards the tumor site, minimizing the concentration of chemotherapeutic agents, radioelements, or toxin used and, therefore, minimizing also the secondary effects of these drugs. In addition, the antibodies can be genetically modified to generate second- and third-generation antibodies with enhanced effector functions.

Within this screening process, we will ascertain additive or synergistic effects of the antibodies with chemotherapeutic drugs, small molecules that inhibit molecular interactions or enzymatic activity of proteins involved in cell signaling, or inhibitors of protein kinases over-expressed in tumor cells. The use of bispecific antibodies in these combinations, although not discarded a priori, will be less relevant since each antibody will recognize tumor surface antigens and functionally decrease tumor size. Thus, the need for bispecific antibodies recognizing both a tumor antigen and simultaneously cells from the immune system to bring them together seems a priori less relevant for the proposed approach than in other situations.

### 3.6. Challenges

The use of therapeutic antibodies has several limitations. Firstly, it seems that the entry of antibodies and immunoconjugates into solid tumors is poor [152]. The concentration of therapeutic antibodies used for cancer treatments are around 100 µg/mL in blood, concentrations believed to be high enough to enable the antibody to reach all the cells within a solid tumor mass [153]. Many examples of solid tumors successfully treated with therapeutic antibodies are available, including HER2^+^ breast tumors treatment with trastuzumab [82,97,98,99]. Secondly, most cancer-specific antigens used as targets of antibodies shed from the cell surface at varying rates and by different mechanisms are still poorly understood (including constitutive and regulated shedding) [153]. The assumption was that antigen shedding leads to a decrease in the amount of antigen present in the tumor cells, concomitant with an increase in circulating antigen, resulting in reduced efficacy of the therapeutic agents (i.e., decreased response to therapeutic antibodies or to immunoconjugate treatments) [153]. Since antibody concentrations for cancer treatment are high, it was assumed that soluble antigens might not represent a significant factor for antibody neutralization [153]. Immunoconjugates are used at much lower therapeutic concentrations due to their toxic effects on normal cells. Therefore, in this case, soluble antigen levels can be high enough to interfere with the action of the immunoconjugates [154,155]. In addition, recent analyses including mathematical models suggest that antigen shedding, rather than reducing the efficacy of the therapeutic antibodies, can significantly improve their efficacy [156].

Thirdly, expression levels of proteins, including cell surface proteins, can change widely during disease progression [153], leading in some cases to the loss of expression of the antigens recognized by the therapeutic antibodies, resulting in a lack of therapeutic response and tumor relapse. Instead of using a single therapeutic antibody, if a combination of antibodies against different cell surface antigens is used, as proposed here, it seems unlikely that all the cell surface antigens recognized by the different antibodies simultaneously will stop being expressed. Therefore, the antibodies recognizing antigens still expressed by the tumor cells will remain effective in killing them.

Fourthly, there is the question of a possible increase in toxicity of combinations of therapeutic antibodies. In general, secondary effects of the therapeutic antibodies, including toxicity, depend to a great extent on the expression pattern of the antigen [5]. Secondary effects are higher for antigens broadly expressed in different cell types, whereas they seem to be very low for antigens with highly restricted expression (i.e., CCR9). They will also depend on antigen density (expressed antigen molecules per cell). Thus, in this context, it is relevant to distinguish between combinations containing immune checkpoint antibodies from combinations of antibodies recognizing distinct tumor over-expressed proteins. Despite their high effectiveness, immune checkpoint antibodies also present important adverse effects [157,158]. Presumably, combinations containing anti-immune checkpoint antibodies will also have relevant secondary effects. The combinations we have discussed in the present manuscript, a priori, will not contain immune checkpoint antibodies but rather antibodies recognizing distinct tumor over-expressed proteins with a restricted expression. Therefore, their secondary effects and toxicity are expected to be much lower than combinations using immune checkpoint antibodies. This seems also to be true for other combinations [159], including combinations with chemotherapeutic agents, where the concentrations of antibodies and chemotherapeutics can be decreased, decreasing the toxicity of such combinations [83].

Fifthly, the immune system has evolved to generate polyclonal responses to an antigen, optimizing in this way its ability to fight disease. Currently, a monoclonal strategy is being used, which might be less efficient than an oligoclonal or polyclonal strategy [29], which will represent an increase in the efficiency of therapeutic responses. In this context, it has also been suggested that analysis of the patient’s genome might lead to the development of new antibodies for cancer treatment, although as far as we are aware, none of these antibodies has made it yet to advanced clinical trials.

## 4. The Goal

The experimental goal is to generate a large panel of therapeutic antibodies that on patients’ biopsies will allow us to pinpoint the antibodies from the panel recognizing antigens expressed by the tumor and then determine the best combination of antibodies to use as initial treatment; it will also allow us to select antibodies for a secondary treatment if required.

Currently, ninety-one monoclonal antibodies have been approved by the FDA, EMA, or other agencies (October 2023), and many others are in process. The approved antibodies have been generated by different laboratories. If the aim is to burst the potential of precision medicine, this requires a strategy change, where large multidisciplinary consortiums with appropriate funding tackle the generation of antibodies against a given type of cancer. The consortiums must be multidisciplinary and require expert bioinformaticians able to identify the patterns of the membrane proteins of interest, with a restricted expression in normal cells and tissues but over-expressed in the tumor type of interest. Medical personnel experienced in the oncological treatments for that tumor type and with access to primary patient samples that can be provided to the consortium and molecular biologists able to generate the required cDNA constructs subcloned in an appropriate expression vector for the immunizations, molecular analyses of the xenotransplanted tumors, and designs for the humanization of the antibodies are required. Immunologists will also be required to participate in the immunization, selection of the antibodies, generation of the xenotransplants, treatments of the xenotransplanted animals with the antibodies, and analysis of the data. These scientists together will help to make the appropriate decisions on which antibody leads to pursue and which ones to discard. Of course, biotechnology and/or pharmaceutical companies might also be part of the consortium since they will be involved at some time in the development of the antibodies (i.e., clinical phases).

This effort might lead to the generation of a relatively large set of antibodies with therapeutic potential that will be screened for each patient’s sample (Figure 6A), where the expectations would be to find several antibodies that positively recognize the tumor cells of a given patient (between four and six on the hypothetical shown in Figure 6), whereas there are antibodies that do not recognize any of these samples and others recognize one tumor sample, two tumor samples, or even three tumor samples (Figure 6B). The screening of a large panel of antibodies with each tumor sample will allow to design initial treatment combinations containing at least two–three antibodies for each patient, which should allow to kill a maximum number of tumor cells, and have a backup or additional combinations of antibodies with chemotherapeutics, inhibitors, etc., in particular in cases of synergy that would be used in patients with adverse secondary effects from the initial treatment or during relapse.

We envisage here that the proposed strategy changes will represent a key burst in personalized/precision medicine for cancer treatment, in particular since the first steps are designed to identify target genes with a restricted expression on normal tissues. Antibodies have been used for pancreatic cancer diagnostics since the middle of the 1980s [160], and in vitro and in vivo experiments to identify possible therapeutic antibodies have been carried out since the beginning of the 1990s [161,162,163,164,165,166,167,168,169,170,171,172,173,174,175,176,177]. These include antibodies inhibiting tumor neo-vascularization [162,175,176], antigens expressed in the tumor cells [161,162,163,164,166,167,168,169,170,171,172,173,175,177], or anti-immune checkpoint antibodies [161,165,174]. However, the only antibodies that have made it to the clinic for the treatment of pancreatic cancer, so far, are the anti-immune checkpoint antibodies, in particular the PD-1 inhibitor (Pembrolizumab). We believe that one of the main reasons for the lack of therapeutic antibodies for pancreatic adenocarcinoma treatment is that although analyses to determine differentially expressed genes in pancreatic cancer have been carried out successfully [178], most of the antibodies tested recognize proteins widely expressed in other cell types, including ADAM-17 [167], CD44 [169], CD47 [171], or CD24 [179].

## 5. Concluding Remarks

The discussed outlined strategy might lead to an important burst of personalized cancer treatments. The main point is that it cannot be carried out by any laboratory on its own; it requires a multidisciplinary team to carry it out, together with collaboration with industry, which at some time or other will be involved. Close collaboration between the different teams might lead to a high chance of success, and the antibodies generated for the treatment of one type of tumor can also be used for the screening of other types since, as shown in the HER2 example, breast tumors that do not belong to this subtype or tumors from other types can also be positive for its expression and could be treated with trastuzumab. Furthermore, it is important to note that all patients with high expression levels of a given antigen, independent of the cancer type or subtype to which they are classified, can benefit from targeted therapy with the appropriate antibody. For example, all patients with high levels of HER2 can benefit from therapy with trastuzumab/pertuzumab independently if their tumors have been classified as mammary HER2^+^, mammary HER2^−^, or other types of tumors, such as colon, lung, or ovarian carcinomas. In addition, this strategy could also be applied to the generation of therapeutic antibodies in other diseases, such as infectious or autoimmune diseases.

## Figures and Tables

**Figure 1 cells-12-02837-f001:**
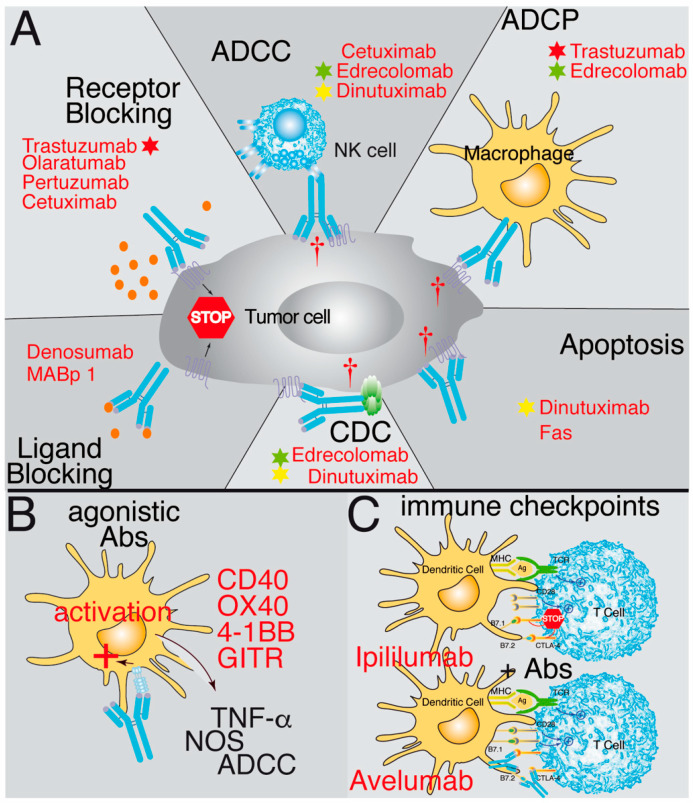
Schematic representation illustrating the mechanisms of action of therapeutic antibodies. (**A**) Some antibodies block the interaction between ligands and receptors by binding to either the ligand (ligand blocking) or the receptor (receptor blocking), preventing the signaling promoting tumor growth. Other antibodies bind to tumor antigens and then are recognized by natural killer cells (NK), triggering their cytotoxic activity, known as ADCC (antibody-dependent cell cytotoxicity). Another possibility is that when the antibody binds to the tumor antigen, it opsonizes the cell and activates phagocytic cells, thereby triggering antibody-dependent cell phagocytosis (ADCP). Additionally, the antibody can fix complement and trigger complement-dependent cytotoxicity (CDC) after binding to the tumor cell. Furthermore, some antibodies can trigger direct apoptosis after binding to an antigen on the tumor’s cell surface. (**B**) Other antibodies have agonistic effects. They identify antigens in the antigen-presenting cells and trigger an activating response of these cells similar to the ligand binding. (**C**) The last group corresponds to antibodies recognizing immune checkpoint antigens, such as CTLA-4, PD-1, or PD-L1. Here, the antibody acts by inhibiting the binding of the ligand and prevents the negative signaling through these receptors. Examples of antibodies functioning through these mechanisms are in red. Stars of different colors indicate antibodies that work through several mechanisms of action.

**Figure 2 cells-12-02837-f002:**
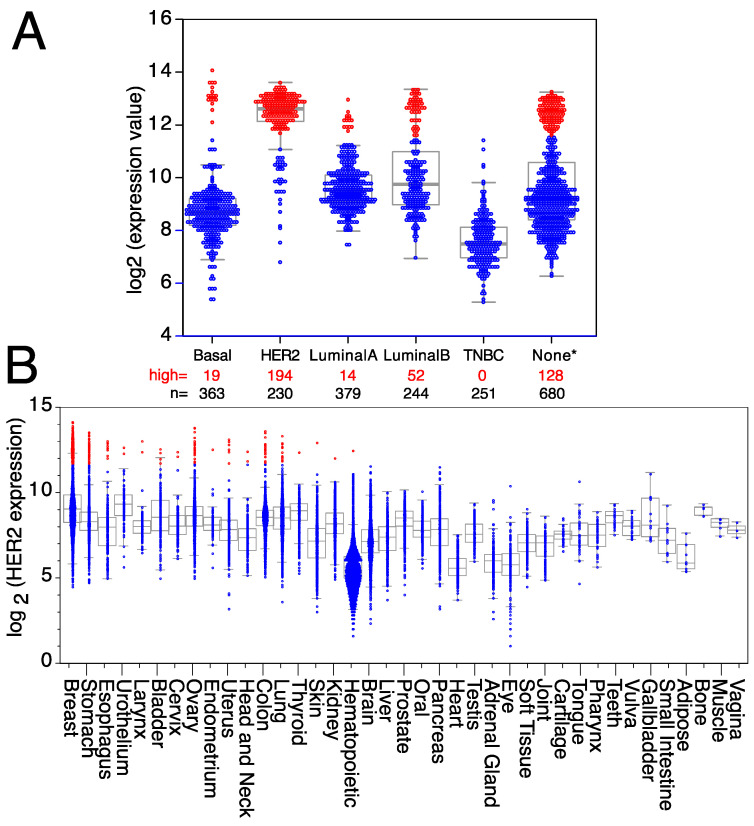
Expression levels of HER2 in breast cancer and other tumor types. Expression levels of HER2 mRNA in (**A**) different breast cancer subtypes (TNBC = triple-negative breast cancer; None* = breast tumors with no data available on subtype). (**B**) Tumors expressing high levels of HER2 are in red, whereas tumors expressing low/medium levels of HER2 are in blue. Medians, the first and third quartiles (boxes), and the 10th and 90th percentiles (whiskers) are indicated for each type of tumor. The number of samples for each tumor type (n) is shown in Appendix A. Expression data were obtained from the GENT2 public database [109].

**Figure 3 cells-12-02837-f003:**
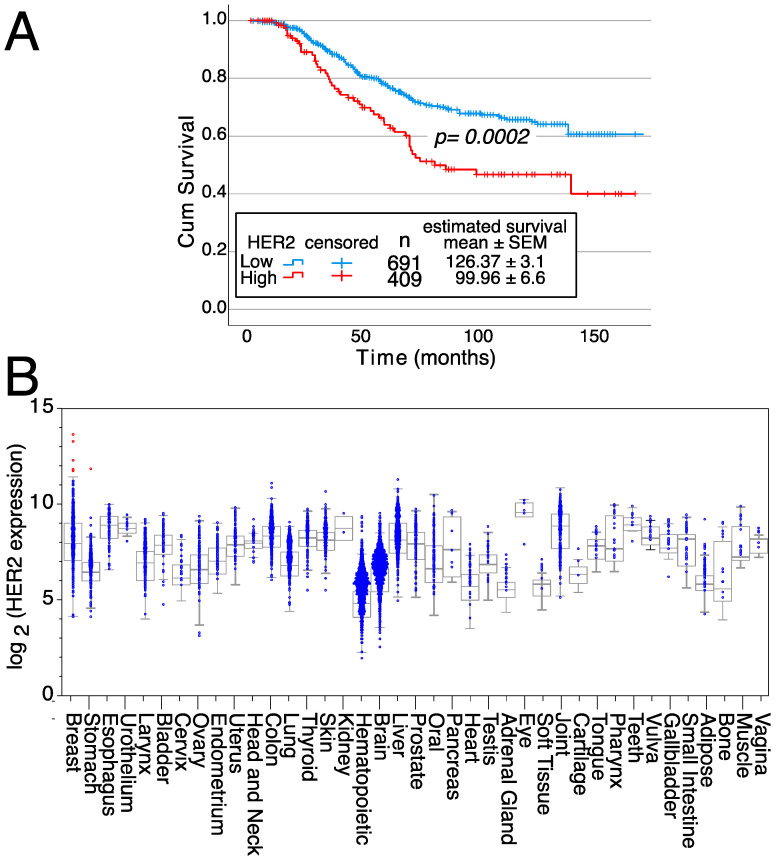
Effects of HER2 on survival and expression in normal tissues. (**A**) Kaplan–Meier survival curves depending on either high or low/intermediate HER2 expression levels, indicating the mean survival ± SEM as well as the statistical significance of the differences in survival curves determined using the Chi-square test. Data from all breast cancer tumor patients from Figure 2A, for which survival data were available, were analyzed. (**B**) HER2 expression levels in normal tissue samples. Samples expressing high levels of HER2 are in red, whereas samples expressing low/medium levels of HER2 are in blue. Medians, the first and third quartiles (boxes), and the 10th and 90th percentiles (whiskers) are shown for each normal tissue. The number of samples for each normal tissue (n) is shown in Appendix A. Expression data were obtained from the GENT2 public database [109].

**Figure 4 cells-12-02837-f004:**
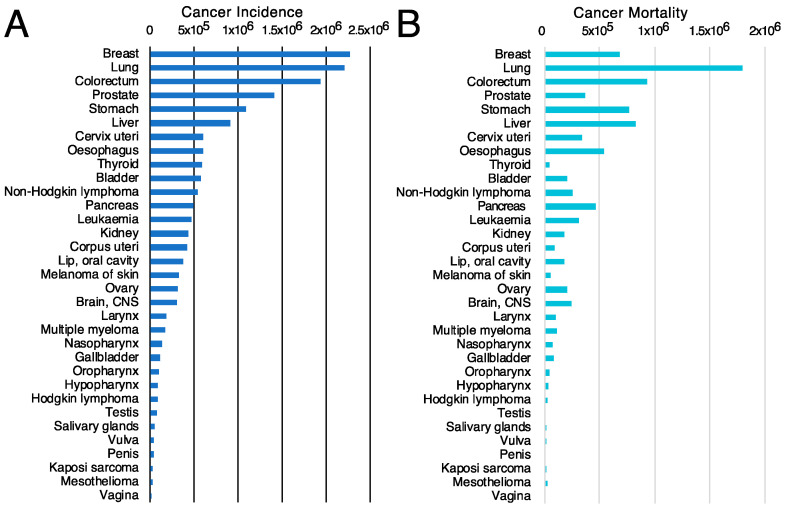
Cancer incidence and cancer mortality worldwide. (**A**) cancer incidence [117] and (**B**) cancer mortality [117] data obtained from GLOBOCAN2020 from the International Agency for Research on Cancer from the World Health Organization [118].

**Figure 5 cells-12-02837-f005:**
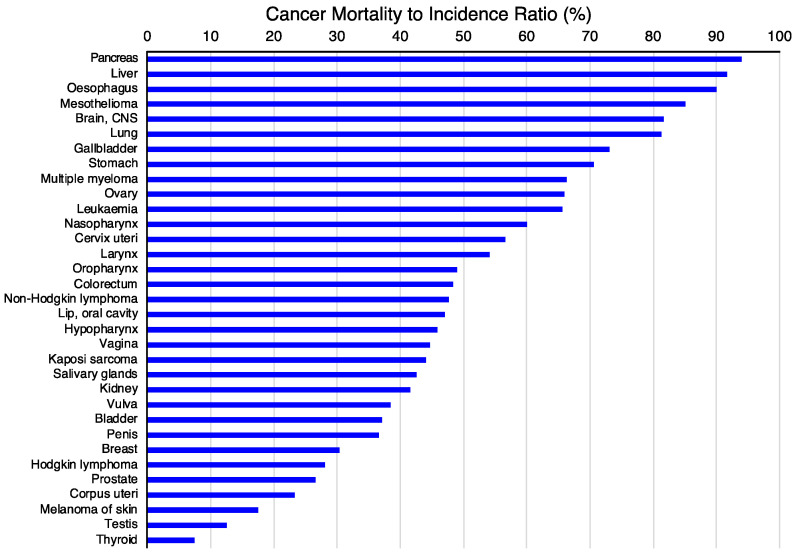
Cancer mortality-to-incidence ratio (MIR). The mortality-to-incidence ratio was calculated from the GLOBOCAN2020 data (presented in Figure 4) by calculating 100 times the mortality-to-incidence ratio, expressed as a percentage [121]. The data were obtained from the International Agency for Research on Cancer from the World Health Organization [118].

**Figure 6 cells-12-02837-f006:**
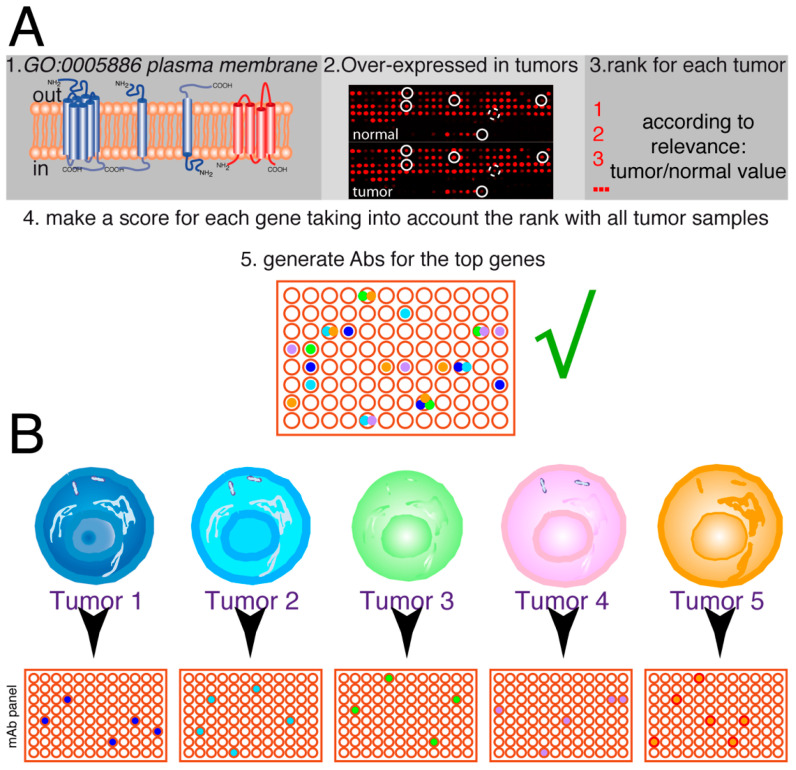
Future of personalized medicine using therapeutic antibodies. (**A**) The idea is to select a series of genes from the human genome, coding for plasma membrane proteins, which are over-expressed in tumors (≥2-fold higher expression levels). On each tumor sample, the tumor/normal expression ratio for each gene allows it to be ranked. Afterward, the scores for the different tumors are analyzed together to make a rank for the genes on that set of tumors. According to the biological features of these genes, the top-ranking ones are used to select mAbs. Then, they are tested for the ability to inhibit or delay tumor growth. This panel of antibodies (in this hypothetical example, represented by 96 different antibodies) will have therapeutic potential for one or more types of tumors. (**B**) The tumor samples for each patient (tumors 1–5) are screened for the expression of the antigens recognized by the antibodies, allowing to select a series of antibodies, positive for each particular tumor (4 to 6 in the hypothetical example, with some antibodies not recognizing any tumor samples and others recognizing one, two, or even three tumor samples). Thus, a combination containing 2–3 different antibodies could be used for the initial treatment of each patient, allowing the killing of the maximum number of tumor cells from the beginning of the treatment.

**Table 1 cells-12-02837-t001:** Distribution of the approved antibodies in therapeutic areas.

Therapeutic Area	Number of Abs
Cancer	91
Immune-mediated disorders	53
Infectious diseases	17
Cardiovascular/hemostasis	10
Metabolic disorders	8
Neurological disorders	7
Ophthalmology	4
Genetic diseases	3
Musculoskeletal disorders	3
Hemostasis	1

## Data Availability

Not applicable.

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
