# Peer review of "Therapeutic Monoclonal Antibodies against Cancer: Present and Future"

_cells, 2023, doi:10.3390/cells12242837_

Round 1

Reviewer 1 Report (Previous Reviewer 2)

Comments and Suggestions for Authors

Issues of the manuscript have been extensively revised. 

Comments on the Quality of English Language

Minor editing of English language is required

Author Response

the manuscript has been edited for English language 

Reviewer 2 Report (Previous Reviewer 3)

Comments and Suggestions for Authors

The manuscript has been improved greatly, however, some points have to be fixed.

1)      As has been previously requested, adding an appropriate graphical abstract transfers the knowledge to the audience.

2)      I suggest the authors present the details of mechanistic Target-Ab action for some candidates.  The authors could use figures for better illustration of this purpose. They performed this as an example in section 1.1 (line 152) but it is not well described.

3)      I strongly suggest the authors draw a table to illustrate the clinically approved antibodies, along with their target and respected cancer.

4)      The quality of Figure 5 is extremely poor. This has not been improved.

5)      Hybridoma technology produces hybridoma to produce mAbs. Unfortunately, the authors did not discuss it in the 1.2.4 section.  

6)      Antibody therapy almost combines with chemotherapy. Why it has not been discussed in this review?

7) The production of monoclonal antibodies based on individual genetic material is one of the goals of future personalized medicine. This is not discussed in this review. 

Author Response

Reviewer 2:

Comments and Suggestions for Authors:

The manuscript has been improved greatly, however, some points have to be fixed.

  • As has been previously requested, adding an appropriate graphical abstract transfers the knowledge to the audience.
  • We have added a graphical abstract as requested
  • I suggest the authors present the details of mechanistic Target-Ab action for some candidates.  The authors could use figures for better illustration of this purpose. They performed this as an example in section 1.1 (line 152) but it is not well described.
  • We have added a figure where the mechanisms of action of mAbs on the tumor cells are described with some antibody examples for each mechanism
  • I strongly suggest the authors draw a table to illustrate the clinically approved antibodies, along with their target and respected cancer.
  • A table to illustrate the clinically approved antibodies, along with their targets and type of cancer approved for was already added as supplemental table I and is maintained in this second revised version. The reason for adding it as a supplementary table is that it contains 91 monoclonal antibodies (the clinically approved for cáncer treatment only) and it spans 6 pages with the table formatting. After asking the editor if it was necessary to add it to the main text or if it was fine as supplementary material she suggested that I could go either way but explain to you the reasons in my reply(supplementary material considering that it occupies 6 pages).
  • The quality of Figure 5 is extremely poor. This has not been improved.
  • Here I have changed the inserted file for a TIFF of 300 dpi (as suggested by the editor). I am sorry for the misunderstanding on the previous revision.
  • Hybridoma technology produces hybridoma to produce mAbs. Unfortunately, the authors did not discuss it in the 1.2.4 section.
  • We have added the explanation in lines 311-315 of the second revised version “Upon immunization of an animal with an antigen, the B cells that produce antibodies recognizing that particular antigen become activated, proliferate, increase their frequency, and facilitate their selection. The hybridoma technology allows the immortalization of the immunoglobulin-secreting B cells, producing large amounts of the desired antibody over time.”
  • Antibody therapy almost combines with chemotherapy. Why it has not been discussed in this review?
  • The combination of antibody therapy with chemotherapy is discussed in lines 139-145 in general and with an example of an anti-CCR9 antibody in combination with vincristine of the R1 version. In the present R2 version, a series of publications showing synergistic effects of Abs with chemotherapy, inhibitors, and dendritic cell vaccines have been added. I agree that the possibility of combining both types of therapies should be mentioned in the text. Hopefully now is clear
  • The production of monoclonal antibodies based on individual genetic material is one of the goals of future personalized medicine. This is not discussed in this review.
  • It is mentioned in lines 441-444 from the revised version 1 “In this context it has also been suggested that analysis of the patient's genome might lead to the development of new antibodies for cancer treatment, although as far as we are aware, none of these antibodies has made it yet to advanced clinical trials.” because as you mention is one of the goals of future personalized medicine. Our point of view is that this will be very useful for antibodies against neoantigens, but over-expressed antigens will make personalized medicine a lot more expensive, will represent a delay in the starting of the treatment, and is not a must.

Reviewer 3 Report (Previous Reviewer 1)

Comments and Suggestions for Authors

The authors have revised their manuscript and it is much improved. A very nice study.

Author Response

There were no requests from this referee

Round 2

Reviewer 2 Report (Previous Reviewer 3)

Comments and Suggestions for Authors

The manuscript has been improved greatly. The graphical abstract is fine. I recommend publishing. 

This manuscript is a resubmission of an earlier submission. The following is a list of the peer review reports and author responses from that submission.

Round 1

Reviewer 1 Report

Comments and Suggestions for Authors

This is an interesting and comprehensive review of antibodies developed for cancer therapy. The authors present an interesting overview and opinion about the directon of precision therapeutics looking to the future. The review also includes very nice illustrations some mining publicly available datasets. These complement and enhance the narrative.

Some areas for development to further improve this nice manuscript are:

1. Two aspects of antibody therapeutics not addressed in the review relate to:

a) agonistic antibodies such as those recognising CD40 and their modes of action and promise for cancer therapy and

b) antibodies with modified Fc regions, such as defucosylated IgG1 antibodies or antibodies of different classes such as IgA, IgE and IgM which may applied for the treatment of cancer in different contexts.

These could also be mentioned in the review.

2. Figure 2A, where the authors discuss that breast tumors with high HER2 mRNA expression levels have a 26% reduced survival, please state the numbers of years follow up for these statistics.

3. Should one conclusion from section 1.1 be that high HER2 expression levels are found across subtypes of breast and other cancer types, therefore selection of patients who might benefit from targeted anti-HER2 therapies should be based on expression of the antigen rather than subype of cancer alone?

4. Please revise the following sentences to improve clarity:

In the abstract: "Here we discuss our view on how these strategic changes might go...": a better verb could be used, such as "might be applied" or"might be implemented"?

"If the ligand-receptor interaction is required for tumor cells survival..."

"For this purpose, it might be useful the incidence (Figure 3A) and 154 mortality (Figure 3B) rates for different types of tumors [52] or the mortality-to-incidence ratio of each type of tumor (Figure 4), which might seem more appropriate since it shows, for example, that thyroid tumors despite being relatively frequent (>5x105 new cases/year, 9th on the incidence list), is the less life threatening type of tumor (>90% survivals), whereas pancreatic cancer, with a similar incidence (~5x105 new cases/year, 12th on the incidence list), is the more life-threatening type of tumor (>90% deaths)."

"Another group recognizes surface antigens in cells of the immune system, either to attract them to the tumor, or even to switch the response of these cells from an immunosuppressed antitumoral status, to another that allow them to effectively attack the tumor [10]."

Comments on the Quality of English Language

Some minor editing is recommended. I have made some suggestions to the authors, however, a review of the language might also be beneficial.

Reviewer 2 Report

Comments and Suggestions for Authors

In this manuscript, Delgado et al summarized their view on new strategies of monoclonal therapeutic antibodies for cancer. This is an important topic, and some key points have been discussed. However, more content should be comprehensively reviewed. 

1. The title should change into "Therapeutic monoclonal antibody", since bispecific antibody, antibody-drug conjugate and other antibody-based therapies have not been involved. 

2. One of the most important parts of monoclonal antibody is the mechanism of drug resistant and strategies to overcome, which have not been discussed in current manuscript. 

3.  Engineering is also an important direction for monoclonal antibody development. Please discuss the efforts that have been made on monoclonal antibody. 

4. Immune checkpoint blocking antibodies and agonist antibodies should be comprehensively discussed. 

5. Tables are required to summarize current targets, drugs and clinical trials for each cancer type.  

6. Present studies were not discussed and cited sufficiently. 

Comments on the Quality of English Language

English should be improved by correcting the grammar issues. 

Reviewer 3 Report

Comments and Suggestions for Authors

1) The abstract must be revised. The authors described a short background about monoclonal antibodies, then wrote their aims of the review, however, they did not mention any latest findings in this area. If they provide numerical findings, the abstract becomes more enriched.

2) Adding an appropriate graphical abstract at the end of the introduction, absorb more audience.

3) I suggest the authors provide a table containing commercially available therapeutic antibodies either FDA-approved or not.

4) Usually antibody therapy is combined with chemotherapy. The authors did not mention anything about it.

5) The quality of Figure 5 is very poor. This must be improved.

6) The authors should write the latest antibodies under research and development for potential immunotherapy. 

7) I suggest the authors add a subsection for describing personalized medicine to develop antibodies from a patient's own genome to treat cancer.

8) I suggest the authors prepare a table for the 1.2.2 section. They need to list cell surface proteins to develop monoclonal therapeutic antibodies.

9) I suggest the authors add a subsection to compare the advantages and disadvantages of both polyclonal and monoclonal antibodies against cancer.

10) I suggest adding therapeutic antibodies to treat various cancers.

Reviewer 4 Report

Comments and Suggestions for Authors

The manuscript describes a historical sequence in the form of a review. It presents in a very detailed way the current fields of application of the different approaches of immunotherapy in a variety of malignant diseases. The field of immunotherapy in malignant diseases has expanded tremendously in recent years. Therefore, not every study could be included in this review. Nevertheless, the authors have provided a well-balanced overview of the most important current applications of immunotherapeutics in this review.

From my point of view, there is no need for further editing of the manuscript at this point, as it would otherwise take up too much space and might interfere with the readability of the manuscript. It should be noted that in future reviews and meta-analyses, individual immunotherapy approaches in defined entities should be considered in more detail. This will provide a better space for critical evaluation.